# Comparison of Nanoparticles and Single-Layer Centrifugation for Separation of Dead from Live Stallion Spermatozoa

**DOI:** 10.3390/vetsci11070307

**Published:** 2024-07-10

**Authors:** Christian Bisiau, Paula Moffett, James Graham, Patrick McCue

**Affiliations:** 1Department of Clinical Sciences, College of Veterinary Medicine and Biomedical Sciences, Colorado State University, Fort Collins, CO 80523, USA; 2Department of Biomedical Sciences, College of Veterinary Medicine and Biomedical Sciences, Colorado State University, Fort Collins, CO 80523, USA; paula.moffett@colostate.edu (P.M.); james.k.graham@colostate.edu (J.G.)

**Keywords:** equine, stallion, spermatozoa, nanoparticles, centrifugation

## Abstract

**Simple Summary:**

In this research project, we wanted to test the use of nanoparticles as an effective method for sorting dead from live spermatozoa from a stallion ejaculate. We compared the use of nanoparticles versus centrifugation with EquiPure™, considered as the gold standard method for obtaining a better quality ejaculate after processing. When we compared the results obtained after processing the samples, centrifugation with EquiPure™ yielded significantly better results than using nanoparticles when analyzed for sperm motility, plasma membrane integrity and sperm acrosome status.

**Abstract:**

The goal of this study was to compare the efficacy of coated iron-core nanoparticles and single-layer centrifugation for separation of dead from live stallion spermatozoa. Our hypothesis was that nanoparticles would bind to dead sperm and allow for separation from live sperm using a magnet, resulting in a population of spermatozoa with a high percentage of total and progressive motility. Treatment Group 1 was an untreated control. Treatment Group 2 (nanoparticles, NP) utilized sperm incubated with nanoparticles followed by application of a magnet to remove dead sperm adhered to the coated nanoparticles. Treatment Group 3 (single-layer centrifugation, SLC) layered sperm above EquiPure™ followed by centrifugation. Semen samples were subsequently evaluated for sperm motility parameters, plasma membrane integrity, acrosome status, and morphology. The SLC technique yielded higher (*p* < 0.05) progressive motility (76 ± 9.2%) than the NP separation technique (59 ± 12.2%) or the untreated control (47.3 ± 5.1%). However, the total number of sperm recovered was higher (*p* < 0.05) in the NP technique (526.2 ± 96.6 × 10^6^) than the SLC procedure (211.7 ± 70 × 10^6^), yielding a higher total number of progressively motile sperm (317.6 ± 109 × 10^6^) recovered using the NP technique than the SLC technique (157.8 ± 43.6 × 10^6^). The percentage of live, acrosome intact sperm recovered was higher for SLC than NP. In summary, the SLC technique yielded a higher percentage of sperm motility, intact plasma membranes, and acrosome integrity, but yielded lower total sperm than with the nanoparticle separation technique.

## 1. Introduction

Artificial insemination with fresh, cooled, or frozen semen is common in the equine breeding industry [1]. Artificial insemination involves semen collection using an artificial vagina, pharmacological *ex copula* ejaculation or by use of a condom [2]. Unfortunately, the ejaculate of some stallions contains a high percentage of immotile or non-viable spermatozoa [3]. Semen with a low percentage of live, motile spermatozoa may be associated with reduced pregnancy rate per cycle. Poor quality semen also limits the number of mares that can be inseminated with each ejaculate [3,4].

Damaged sperm can produce reactive oxygen species (ROS) that adversely affect survival of live spermatozoa [5]. If the ejaculate is of poor quality, it is recommended that dead sperm be separated from live spermatozoa prior to fresh insemination or cooled storage. Techniques for live–dead separation include a swim-up technique [6], density gradient centrifugation [7], single-layer centrifugation [8], and filtration through columns [9]. An alternative method of sperm separation involves targeting specific markers of sperm quality, such as defects in the acrosome and/or plasma membrane using nanoparticles coated with specific probes [10,11]. Nanoparticles with an iron-core can be coated with probes such as peanut and pea agglutinin lectins (PNA/PSA), annexin V, or anti-ubiquitin antibodies, and the nanoparticle-bound spermatozoa can be separated from non-bound spermatozoa using a magnet [12,13]. The nanoparticle separation technique has been reported to be moderately successful in sorting dead from live sperm in humans [14], boars [10], bulls [15], stallions [16,17], donkeys [18], and camels [19].

The objective of the present study was to determine the efficacy of nanoparticles in the separation of dead from live stallion spermatozoa. The study model utilized a mixture of fresh ejaculated live motile spermatozoa to which a specific number of snap frozen killed spermatozoa was added. Our hypothesis was that nanoparticles coated with a probe would bind to dead sperm and allow for subsequent separation from live sperm using a magnet, resulting in a population of spermatozoa with an increased percentage of total and progressive motility.

## 2. Materials and Methods

### 2.1. Animals and Semen Collection

One ejaculate of semen was collected from each of 6 stallions, including 5 Quarter Horses and 1 Irish Draft, using a Colorado model artificial vagina (Animal Reproduction Systems, Chino, CA, USA) with an in-line nylon gel filter (Animal Reproduction Systems, Chino, CA, USA).

### 2.2. Media

*TALP-E*: Equine TALP was prepared with NaPyruvate (0.0022 g), NaLactate (0.368 g), Glucose (0.090 g), HEPES (0.238 g), and BSA (0.3 g) diluted to 100 mL with Tyrodes solution [(NaCl (0.569 g), KCl (0.023 g), Na_2_HPO_4_ (0.004 g), NaHCO_3_ (0.209 g), CaCl-2H_2_O (0.029 g), and MgCl_2_-6H_2_O (0.008 g)].

Semen Extender: INRA 96 equine semen extender (IMV Technologies, L’Aigle, Basse-Normandie, France) was used; INRA 96 contains a purified fraction of milk micellar proteins plus antibiotics (penicillin, 0.038 mg/mL and gentamicin, 0.105 mg/mL) and an antifungal agent (amphotericin B, 0.315 µg/mL).

Colloid: Silane-coated silica colloid in a buffered salt solution (EquiPure™, Nidacon International AB, Mölndal, Sweden) was used for the single-layer centrifugation.

Nanoparticles (ULTRAsep™): Nanoparticles with an iron-core coated with a proprietary probe designed to bind to dead spermatozoa was provided by ST Genetics (Navasota, TX, USA). Ten mL of INRA 96 extender was added to the concentrated nanoparticle solution to make the nanoparticle working solution.

### 2.3. Semen Evaluation

Volume: The volume of the gel-free fraction of the original ejaculate, as well as the volume of the supernatant after nanoparticle separation was assessed using a graduated cylinder. The volume of the gel fraction was not measured and discarded.

Concentration: The concentration of spermatozoa in the original ejaculate, as well as the sperm concentration in the supernatant, the remnant, and the extended pellet fractions were measured using a NucleoCounter SP-100^®^ (ChemoMetec, Allerød, Denmark).

Sperm Motility: Total and progressive sperm motility was assessed using a computer-assisted sperm analysis (CASA) system (SpermVision^®^; Minitube Verona, WI, USA). Initial sperm motility was assessed by diluting sperm to a concentration of 25 million sperm/mL. A calibrated pipette, glass slides, and cover slip were used.

Sperm Morphology: Sperm morphology was evaluated using a Differential Interference Contrast (DIC) microscope (Olympus BX51, Olympus Global, Center Valley, CA, USA) at 100× magnification (oil immersion). The sperm morphologies were classified as normal or as having head abnormalities, detached heads, midpiece abnormalities, bent tails, coiled tails, proximal droplets, and distal droplets.

Flow-cytometer evaluation: FITC/PNA, PI, and SYTO stains (Sigma-Aldrich; St Louis, MO, USA) were used to evaluate sperm plasma membrane and acrosome status [20]. The emission wavelength for the fluorescent stains were 525 nm, 617 nm, and 621 nm, respectively. During the fluorescent stain evaluation using the flow-cytometer, 10.000 cells were evaluated following the guidelines of BD Accuri™ C6 plus. One replicate per category was evaluated.

### 2.4. Sperm Sorting Procedures

Sperm sorting via single-layer centrifugation (SLC): A total of 600 million sperm was diluted 1:1 (*v*/*v*) with INRA 96 and gently placed on top of 4 mL EquiPure™ in a 15 mL conical tube [21]. The tube was subsequently centrifuged at 300× *g* for 30 min. The sperm pellet was then resuspended in 1 mL of INRA 96 extender prior to evaluation.

Sperm sorting via nanoparticles (NP): A total of 600 million sperm was diluted to a volume of 40 mL and a concentration of 15 million sperm per mL with INRA 96 extender were placed in a 50 mL conical tube. A volume of 2.4 mL of ULTRAsep™ nanoparticle working solution was added to the diluted semen and gently mixed. The mixture was incubated in a light-proof cabinet at room temperature (22 °C) for 15 min. A magnet was then applied to the side of the conical tube for 5 min. The supernatant containing spermatozoa that were not adherent to the iron-core nanoparticles was decanted into a new conical tube, while the remnant, containing spermatozoa adherent to the nanoparticles was held in place by the magnet in the original conical tube.

### 2.5. Experimental Design

Comparison of single-layer centrifugation (SLC) and nanoparticle separation. Ejaculates from 6 stallions were collected and divided into 3 aliquots (Treatment Groups 1, 2, and 3), all of which had half of the sperm population killed via submersion in liquid nitrogen followed by thawing in a water bath at 37 °C. Treatment Group 1 (control) utilized 600 million total sperm, extended with INRA 96 to 15 million sperm per mL in a volume of 40 mL and stored for 20 min in a light-proof cabinet at room temperature (22 °C). Treatment Group 2 (nanoparticles, NP) utilized 600 million total sperm diluted with INRA 96 to a concentration of 15 million sperm per mL in a 40 mL volume. A ratio of 400 µL nanoparticle solution per 100 million sperm (2400 µL volume of nanoparticle solution) was used, and the mixture was incubated for 15 min at 22 °C, followed by a 5 min application of the magnet. Treatment Group 3 (single-layer centrifugation, SLC) was performed by diluting raw semen containing 600 million total sperm 1:1 (*v*/*v*) in INRA 96 extender and layering the extended semen on top of 4 mL EquiPure™ in a 15 mL conical tube. The tube was subsequently centrifuged at 300× *g* for 30 min.

Definitions:SLC pellet: Sperm concentrated at the bottom of the conical tube after centrifugation with EquiPure™, subsequently reconstituted with INRA 96;SLC Supernatant: Sperm remaining suspended in solution above the pellet after centrifugation with EquiPure™;NP Supernatant: Sperm suspended in solution, not bound to nanoparticles;NP Remnant: Sperm bound to nanoparticles and held in the conical tube by the magnet after decanting off the supernatant;Control: Sperm diluted in INRA 96 extender only.

### 2.6. Statistics

Normal distribution was checked, and one-way ANOVA followed by Tukey’s test was performed to compare the results. Data are presented as the mean ± standard deviation (SD). Values were considered to be statistically different at *p* < 0.05.

## 3. Results

Single-layer centrifugation yielded a population of sperm with a significantly (*p* < 0.05) higher total and progressive sperm motility (82.7 ± 8.3% and 76 ± 9.2%, respectively) than either nanoparticle separation (62.3 ± 11.3% and 59 ± 12.2%, respectively) or the untreated control (50.3 ± 6.3% and 47.3 ± 5.1%, respectively) (Table 1 and Figure 1). There was no difference (*p* > 0.05) in sperm motility parameters between the control and nanoparticle treated groups.

The total number of spermatozoa recovered was significantly higher (*p* < 0.05) for the nanoparticle technique (482.5 ± 137.3 million sperm; 80.41% recovery) than the single-layer centrifugation technique (229.8 ± 66.9 million sperm; 38.3% recovery).

Single-layer centrifugation with EquiPure™ yielded a significantly (*p* < 0.05) higher percentage of live, acrosome intact spermatozoa (61 ± 11.1%) than either nanoparticle separation (43.1 ± 3.3%) or the untreated control group (35.5 ± 3.1%). There was no significant difference (*p* > 0.05) between the percentage of live, acrosome intact spermatozoa after nanoparticle separation and the control group (Figure 2).

There was no difference in percentage of morphologically normal sperm recovered after single-layer centrifugation, nanoparticle separation, and the untreated control group (*p* > 0.05) (Figure 3).

## 4. Discussion

Removal of dead spermatozoa from an ejaculate after semen collection has been reported to prolong the viability of the remaining live spermatozoa [5]. Damaged spermatozoa are capable of generating reactive oxygen species (ROS) which play a physiological role in signaling events controlling sperm capacitation [5,22]. Generation of ROS in vitro through the xanthine oxidase system results in a reduction in sperm motility, viability, and sperm–oocyte fusion [23]. Hydrogen peroxidase appears to be the primary ROS responsible for these changes [24].

Laboratory techniques to separate dead from live spermatozoa have been implemented with variable success. Felmer and coworkers compared the use of a swim-up technique and density gradient centrifugation with Percoll™ for separation of bull spermatozoa [6]. The latter technology yielded a higher percentage of total and progressively motile sperm as well as spermatozoa with intact acrosome membranes compared to the swim-up technique. Kikuchi and coworkers compared density gradient centrifugation using Percoll™ versus a simple centrifugation procedure for separation of boar spermatozoa [7]. The Percoll™ separation method enhanced the recovery of sperm with intact plasma and acrosome membranes and increased cleavage and blastocyst rates after in vitro fertilization as compared with the results obtained after simple centrifugation.

Single-layer centrifugation has been reported to preferentially select stallion spermatozoa that are motile, morphologically normal, with intact plasma membranes, and with good chromatin integrity from the rest of the ejaculate using a species-specific colloid [2]. Furthermore, sperm capacitation status was not affected, and sperm maintained their motility, viability, and chromatin integrity for a longer period of time than untreated spermatozoa [25,26].

The ubiquitin system involves a conserved small protein that selectively causes the degradation of specific proteins in eukaryotic cells. In this pathway, proteins are targeted for degradation via covalent ligation to ubiquitin [27]. Sutovsky and others used nanoparticles coated with either a lectin (PNA/PSA) or monoclonal anti-ubiquitin antibodies to separate bull spermatozoa [15]. Sperm viability was significantly increased in samples nanopurified with PNA particles, but were not different using PSA, anti-ubiquitin antibodies, or the untreated control. In an in vitro fertilization trial, fertilization rate was greater using sperm that were separated with anti-ubiquitin nanoparticles than with the lectins PNA/PSA or mixed ubiquitin/PNA nanoparticles. In a field trial, pregnancy rates were significantly increased when cattle were inseminated using sperm that had been sorted with nanoparticles coated with the lectin PNA versus anti-ubiquitin-coated nanoparticles or an untreated control [15].

Spermatozoa have been shown to display characteristics typical of apoptosis such as caspase activation, decreased mitochondrial membrane potential (MMP), and plasma membrane translocation of phosphatidylserine (PS). Annexin V has a high affinity for phosphatidylserine and has been shown to be very selective [28]. Annexin V-coated nanoparticles were used by Grunewald and colleagues (2006) [29] to separate human spermatozoa. After separation, annexin V negative sperm had significantly higher levels of intact mitochondria following cryopreservation and thawing as compared to sperm that were not treated.

Morris and coworkers reported a significant effect of stallion and treatment on total motility, progressive motility, and sperm concentration when using nanoparticles coated with PNA to separate stallion spermatozoa [16]. Processing samples through EquiPure™ gradient significantly increased the proportion of intact acrosomes compared with the untreated control. Nanoparticle treated groups experienced less DNA damage due to oxidative stress when compared to the control and EquiPure™ gradient groups. The researchers concluded that nanoparticle treatment is not detrimental to sperm motility, viability, or oxidation status.

Coutinho da Silva and coworkers noted that annexin V-coated nanoparticles were effective at removal of apoptotic stallion sperm from the ejaculate [17]. However, removal of apoptotic sperm from the ejaculate did not significantly improve sperm parameters (total and progressive sperm motility, apoptotic sperm, live sperm, or morphology) compared with the untreated control.

Sperm recovery after nanoparticle treatment was higher than the one after centrifugation with a colloid due to the fact that fewer dead spermatozoa had been removed from the sample. Hence, more dead/moribund spermatozoa remained in the nanoparticle sample compared with the centrifugation with EquiPure™. One could say that the fact that no centrifugation is needed is an advantage over a centrifugation with EquiPure™. However, the benefit of using this specific proprietary nanoparticle was not high enough to recommend this specific method over a colloid centrifugation.

Surprisingly, sperm morphology did not improve when centrifugation with a colloid was used. This could possibly be due to the fact that sperm were killed via submersion in liquid nitrogen, which is not a physiological way for sperm to die.

Overall, both methods of “purifying” a semen sample are easy to perform and would take the same time to process. However, the nanoparticles can be much more expensive than a commercially available colloid such as EquiPure™.

The present study compared iron-core nanoparticles coated with a proprietary probe and single-layer centrifugation for separation of dead from live stallion spermatozoa. In summary, the SLC technique yielded higher total sperm motility and progressive sperm motility, as well as a higher percentage of plasma membrane and acrosome intact spermatozoa than the nanoparticle technique.

## 5. Conclusions

With this study, we can conclude that the use of iron-core nanoparticles ULTRAsep™ had no statistically significance impact on equine sperm quality as evaluated by analysis of sperm progressive motility and sperm plasma integrity and acrosome status compare to an untreated control. However, when a centrifugation with the commercially available colloid EquiPure™ was used, we were able to find significant improvement in all the previously mentioned characteristics.

Even though a treatment with nanoparticles has the benefit of not needing any centrifugation, which for some practitioners has proven to be cumbersome, the semen quality after such treatment has not yielded better results than the proven method of centrifugation with a colloid such as EquiPure™.

## Figures and Tables

**Figure 1 vetsci-11-00307-f001:**
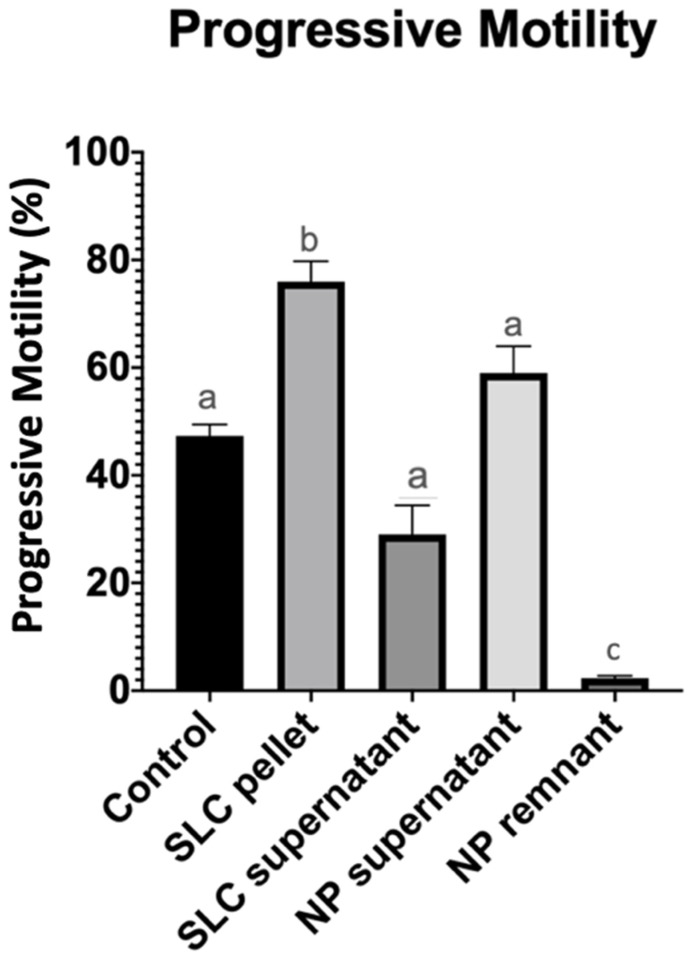
Progressive sperm motility after 20 min of incubation at 22 °C (control group), after single-layer centrifugation (SLC sperm pellet and SLC supernatant), and after nanoparticle separation (NP supernatant and NP remnant). a, b, c—statistically significant.

**Figure 2 vetsci-11-00307-f002:**
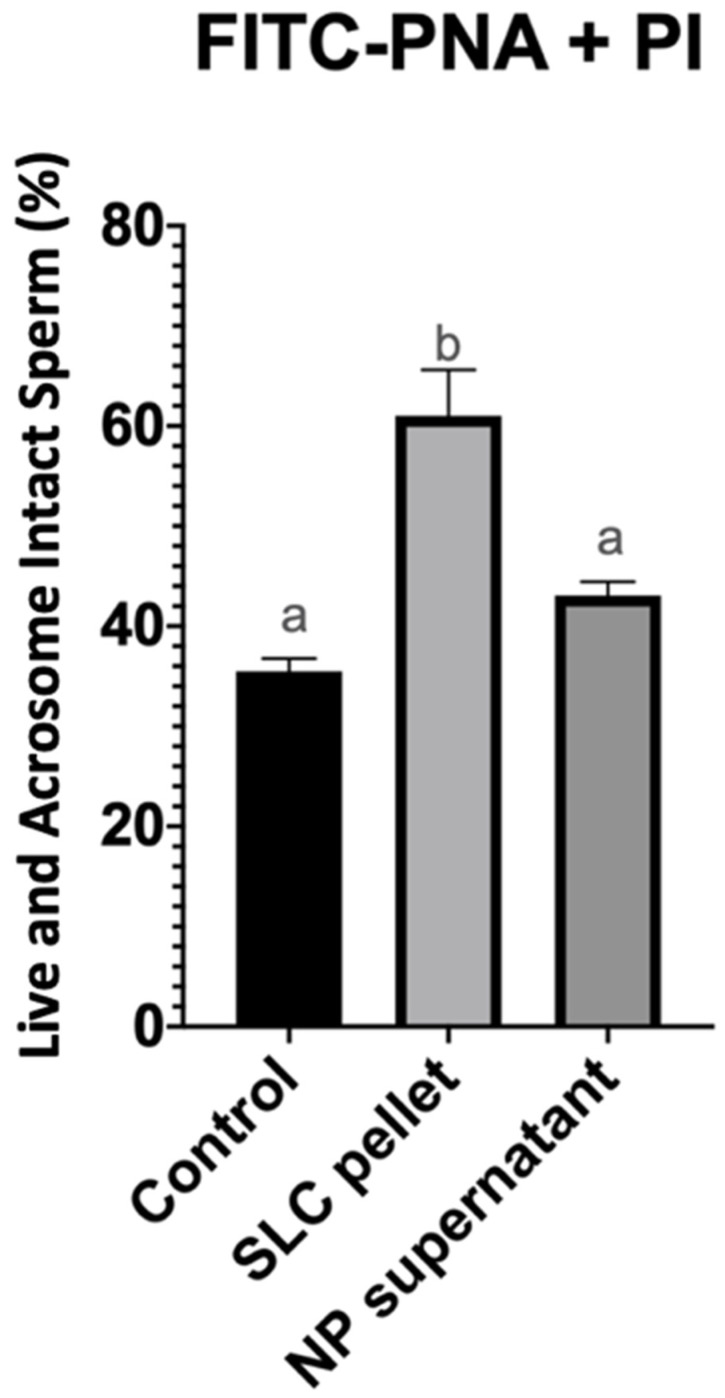
Percentage of live and acrosome intact spermatozoa after 20 min of incubation at 22 °C (control group), after single-layer centrifugation (SLC sperm pellet), and after nanoparticle separation (NP supernatant). a, b—statistically different.

**Figure 3 vetsci-11-00307-f003:**
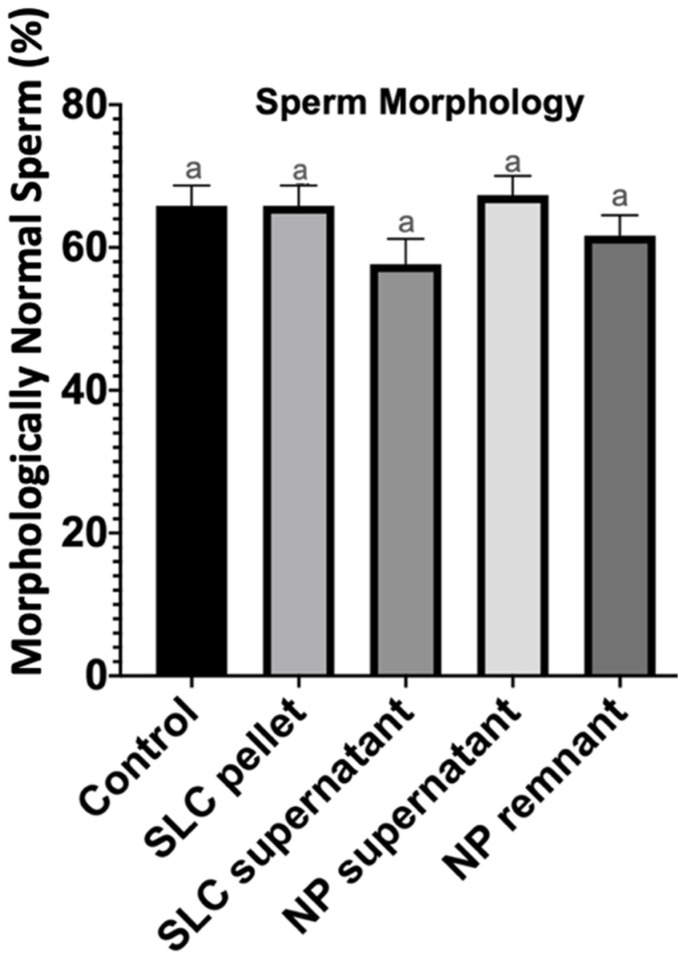
Percentage of morphologically normal spermatozoa after 20 min of incubation at 22 °C (control group), after single-layer centrifugation (SLC sperm pellet and SLC supernatant), and after nanoparticle separation (NP supernatant and NP remnant). a—no statistical differences between columns.

**Table 1 vetsci-11-00307-t001:** Comparison of total and progressive sperm motility and total sperm recovery following nanoparticle (NP) separation and single-layer centrifugation (SLC).

Treatment Groups	Total Motility (%)	Progressive Motility (%)	Total Number of Sperm Recovered (Millions)	Total Number of Progressively Motile Sperm Recovered (Millions)
Control	50.3 ± 6.3 ^a^	47.3 ± 5.1 ^a^	620.2 ± 51.8 ^a^	293.7 ± 41.9 ^a^
NP supernatant	62.3 ± 11.3 ^a^	59 ± 12.2 ^a^	482.5 ± 137.3 ^a^	317.6 ± 109 ^a^
SLC pellet	82.7 ± 8.3 ^b^	76 ± 9.2 ^b^	229.8 ± 66.9 ^b^	157.8 ± 43.6 ^b^

^a,b^ Data in columns with different superscript are significantly different (*p* < 0.05).

## Data Availability

All data are contained within the article.

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
