# Peer review of "Comparison of Nanoparticles and Single-Layer Centrifugation for Separation of Dead from Live Stallion Spermatozoa"

_vetsci, 2024, doi:10.3390/vetsci11070307_

Round 1
Reviewer 1 Report
Comments and Suggestions for Authors
This paper presents a clear and structured methodology that effectively addresses the research question. However, a few editions could improve the manuscript.
The summary could benefit from more comprehensive details that directly highlight the main findings and implications of the study. When referencing "this method" in the text, it would be beneficial to specify which method is being discussed to avoid ambiguity and enhance the reader's understanding of the key techniques employed in the research.
Abstract:
Replace the generic term "nano-particle" with a specific term, such as "ULTRAStep."
Method:
Please include information about the brand and batch number for all chemicals and consumables to enable replication of the experiment. The nano-particles are highly sensitive to pH and osmolarity; thus, having the exact components is critical.
Please specify the molarity of the chemicals used.
Provide the settings for CASA.
Please clarify whether you checked for normal distribution and equal variances of the data prior to using ANOVA.
Results:
It is somewhat interesting that the authors did not observe any differences in the morphology of the semen post-selection even with Androcol. This could be due to the method used for “killing” the sperm cells before adding them to the semen. Please discuss this in the discussion section.
Discussion:
This section needs more discussion about the results, including the advantages and disadvantages of each method. The authors found a higher overall number of spermatozoa with nano-particles (NP). Discuss whether this would be advantageous in specific cases.
Discuss the feasibility of each method in terms of time and equipment needed for use in clinical practice. Since centrifugation might affect some stallion semen, NP might be useful for those cases. Please elaborate on this subjects.
Author Response
Reviewer,
Thank you very much for your constrictive criticism and comments.
We at Colorado State University, USA we use the manufacturer settings which I was not able to find but in a way we know that are accurate. Because we used the same settings for all the stallions and ejaculates and treatments we gathered that it might not matter as much. However, if you feel like we should include them, I can try to do some more digging.
Regarding the nanoparticles UltraSep, they have this proprietary substance coating the iron core that they did not disclose. No pH or osmolarity was disclose nor measure by us either. We applied them and kept them stored as indicated by the manufacturer.
Thank you very much for your comments.
Reviewer 2 Report
Comments and Suggestions for Authors
This nanoparticles (ULTRAsep™) product is new, therefore, the more details are need it.
Author Response
Dear reviewer,
Thank you very much for your comment. You are correct, the product ULTRAsep is relatively new and I wish I could have more things to say about. Unfortunately, the nanoparticles are proprietary and absolutely nothing was disclosed to us. From other nanoparticles that have been use in the past by others, I could assume or infer that most likely a lectin is coating the iron core nanoparticle, but nothing was disclosed, which makes it hard to say much about them.
Thank you very much for reading the article and commenting on it.
Reviewer 3 Report
Comments and Suggestions for Authors
The manuscript vetsci-3047269:Comparison of nanoparticles and single-layer centrifugation for separation of dead from live stallion spermatozoa. Describes the use of no separation, separation via EquiPureTM, and separation via nanoparticles. Results found that the best recovery of progressively motile sperm and viable sperm was via the EquiPure TM procedure. Recovery of the progressively motile and viable sperm was no different between control (no separation) and nanoparticles.
The manuscript is well-written in English. The methods and results are further clearly described. My only negative comment is that in the discussion there was no attempt to say why the results turned out as they did. For example, why was the nanoparticle treatment not better? The discussion focused on results by others but did not address this experiment's results.
Author Response
Dear reviewer,
Your comment is very accurate. Due to the fact that these specific nanoparticles have a proprietary product coating the iron core, that was not disclose to us, or any other aspect of how they work, makes it hard to identify why they did not work. I did not want to speculate or anything similar, therefore I did not write any possible reasons.
Thank you very much for your comment. Please let me know if I can answer anything else for you.